# Kinetic and Kinematic Analysis of Landing during Standing Back Somersault Using Three Technical Arm Swings in Artistic Gymnastics

**DOI:** 10.3390/jfmk8010010

**Published:** 2023-01-13

**Authors:** Bessem Mkaouer, Hounaida Akkari-Ghazouani, Samiha Amara, Raja Bouguezzi, Monèm Jemni, Helmi Chaabene

**Affiliations:** 1Department of Individual Sports, High Institute of Sport and Physical Education of Ksar Said, University of Manouba, Manouba 2010, Tunisia; 2Tunisian Research Laboratory “Sport Performance Optimization”, National Centre of Medicine and Science in Sport, Tunis 1004, Tunisia; 3Department of Physical Education and Sport Sciences, College of Education, Sultan Qaboos University, P.O. Box 50 Al-Khod, Muscat 123, Oman; 4Research Unit (UR17JS01) Sport Performance, Health & Society, Higher Institute of Sport and Physical Education of Ksar Saïd, University of Manouba, Manouba 2010, Tunisia; 5The Carrick Institute of Neuroscience, Cap Canaveral, FL 32920, USA; 6Department of Sports and Health Sciences, Faculty of Human Sciences, University of Potsdam, 14469 Potsdam, Germany

**Keywords:** motion analysis, acrobatics, preparatory phase, landing, stability

## Abstract

The crucial criteria when assessing technical performance in artistic gymnastics is the higher elevation of the gymnast’s body and a stable landing (i.e., stick-landing). The purpose of this study was to compare kinetic and kinematic parameters during the landing phase of standing back somersaults (SBS) following three technical arm-swing performed during the preparatory phase in high-level male gymnasts. The three different arm-swing pertain to three “gymnastics schools”, i.e., Russian, Chinese, and Romanian. Six high-level male gymnasts participated in this study. Three arm-swing with different angles (i.e., SBS_270°_, SBS_180°_, and SBS_90°_) were randomly performed. A 3D kinetic and kinematic analysis was conducted. Results showed significant variation in the landing angle (*p* = 0.009) across the three arm-swing techniques. The SBS_90°_ arm-swing resulted in the closest angle to the vertical. Additionally, the SBS_90°_ arm-swing technique induced the lowest horizontal and vertical force values upon landing compared to the other arm-swing techniques (SBS_270°_: *p* = 0.023 and 0.009, respectively; SBS_180°_: *p* = 0.004 and 0.080, respectively). The same was noted for the horizontal velocity (*p* = 0.021) with the lowest values noted for the SBS_90°_ arm-swing technique. However, the best opening angle was observed during the SBS_270°_ technique, since it presented the best vertical displacement. In conclusion, the SBS with a SBS_90°_ arm-swing seems to favor a better absorption of the ground reaction force upon landing by reducing the intensity of the impact with the ground and by affording a landing angle closer to the vertical in high-level male gymnasts.

## 1. Introduction

Artistic gymnastics (AG) is a judgment sport in which the landing concludes every event/acrobatic series [1]. As such, the final judgment score is determined to a great extent by the quality of the landing [1]. Earlier studies showed a low success rate at landing in AG, with a high error rate reaching 71.9% on floor exercise [2]. There is also evidence indicating that the rate of lower limb injuries is high during the landing phase on the floor exercise (40%) [3,4,5,6].

On the other hand, the preparation phase (i.e., run-up technique), such as during the floor exercise, is crucial for the success and safety of acrobatic elements in AG. In fact, the preparation phase moderates the amount of momentum necessary for the successful performance of acrobatic elements [7]. More particularly, it has been shown that the preparation phase of an acrobatic element could affect its last phase (i.e., landing phase) [8]. In this sense, the actions of the arms, such as during the preparation phase are very important for a successful performance of acrobatic movements during the floor exercise [7,9,10]. In this sense, there is evidence that the inertia generated by the displacement, speed, and/or opening angle of the arms can markedly affect the quality of the acrobatic element as a whole [7,11] and the landing phase more particularly [8].

The standing back somersault (SBS) is a common floor acrobatic exercise. The preparation phase of the SBS can be performed according to three different schools of AG (i.e., Chinese, Romanian, and Russian) [7]. The difference between these schools is mainly related to the action of the arms. In the Russian school, gymnasts start the arms vertically (arm/trunk angle opening is 180°), and then perform a 270° oscillation in the descending phase by lowering the arms from the top to the front and the back to the end in the backward horizontal position [7,12,13,14,15] (Figure 1a). According to the Chinese school, gymnasts begin with arms extended horizontally forward; in the downward phase, they then perform a 180° oscillation by lowering them downwards and backwards to reach a horizontal back position [7,16,17,18,19] (Figure 1b). However, as per the Romanian school, gymnasts start with their arms extended and lowered along the body; they perform a 90° oscillation going backwards, ending in a horizontal back position [7,20,21,22], (Figure 1c).

It is worth noting that several previous studies focused on the landing phase during the SBS [17,24,25,26,27,28]. However, none of them has addressed the effects of different technical arm-swings during the preparation phase on the final phase, which is the landing phase of the SBS. Therefore, this study aimed to compare the kinetic and kinematic parameters during the landing phase of SBS following three different technical arm-swing performed during the preparatory phase in high-level male gymnasts. More specifically, we aimed to identify which of the three arm-swing techniques during the preparatory phase results in a more efficient and stable landing (stick-landing) with a minimum joint impact.

## 2. Materials and Methods

### 2.1. Participants

A priori power analysis with type I error of 0.05 and 80% statistical power was computed using GPower software (Version 3.1, University of Dusseldorf, Dusseldorf, Germany [29]). The analysis indicated that a minimum of six participants are sufficient to observe a significant, large effect size (f = 50) for kinetic (i.e., vertical and horizontal ground reaction force) and kinematic (i.e., joint angles and velocity) variables [7]. Therefore, six senior men’s artistic gymnastics members of the Tunisian national team (age 23.17 ± 1.61 years; height 1.65 ± 0.05 m; body mass 56.80 ± 7.66 kg) agreed to participate in this study. Of note, all gymnasts are familiar with the three arm-swing techniques as they have been trained by Romanian, Russian, and Chinese coaches throughout their careers. The inclusion criteria included (1) to be ranked at an international level with participation in world cups and/or championships, (2) average training volume of 25 ± 2 h per week, (3) being healthy, with no muscular, neurological, or tendon injuries in the last six months, and (4) able to perfectly perform a SBS using the three arm-swing techniques (i.e., SBS_270°_, SBS_180°_, and SBS_90°_). After being informed in advance of the procedures, methods, benefits, and possible risks of the study, each participant had to review and sign a consent form to participate in the study. The experimental protocol was performed per the Declaration of Helsinki for human experimentation [30] and was approved by the Ethical Committee of the National Centre of Medicine and Science in Sport (LR09SEP01).

### 2.2. Experimental Design

The research design was a simultaneous 3D dual approach study (i.e., a dual approach of kinetic and kinematic) of SBS landing using three arm-swing techniques. These three backswings techniques had different arm-swing angles and trajectories during the preparatory phase of take-off: the first was performed with an oscillation/trajectory angle of 270° from the top to the back and then back again to the upward position (SBS_270°_) (Figure 1a); the second was performed with an oscillation/trajectory angle of 180° from the front to the rear and then back again to the front horizontal position (SBS_180°_) (Figure 1b) while the third was performed with 90° oscillation/trajectory angle from the bottom towards the rear and back again to the downward position (SBS_90°_) (Figure 1c).

Kinetic data were measured using a dual Kistler force plate (9281C, sampling frequency 1000 Hz, size 60 × 40 cm, Kistler Instruments, Switzerland) and analyzed using Bioware Performance Analysis Software 5.1.10 (Kistler Instruments, Winterthur, Switzerland). Ground reaction force (F_y_, F_x,_ and F_z_), moments of force (M_y_, M_x,_ and M_z_), and power (P_y_, P_x,_ and P_z_) were analyzed at the moment of landing.

For kinematic data, twenty retro-reflective body markers were recorded using two high-speed cameras NAC (HSV-500C^3^, sampling frequency 500 Hz, NAC Motion Analysis, Corp., Santa Rosa, CA, USA). Body markers were digitized using Movias video-based data analysis system 2.0.4 (NAC Motion Analysis, Corp., Santa Rosa, CA, USA). The body segments’ centers of mass were computed using the Matsui [31] model. The center of mass (COM) velocity (V_y_, V_x,_ and V_z_), the landing angle (∠_L_), and the hip (∠_H_) and knee joints angles (∠_K_) at landing were recorded and analyzed.

### 2.3. Procedures

The experiment took place in the laboratory of the National Centre of Medicine and Science in Sport across three days, starting at 4:00 p.m. to 6:00 p.m. under the following environmental conditions: average temperature of 23 °C and humidity of 35%. Two Kistler force plates were synchronized with two NAC high-speed cameras. Both cameras were placed at 5 m from the center of the force plate, the first one at the front and the second sideways. The peak force and the moment of force (i.e., quantified as the peak slope of theforce-time curve [i.e., Δ_force_/Δ_time_]), upon landing during the SBS were recorded. The kinematic analysis was conducted in three-dimension (3D). A semi-automatic digitalization was used with quantic-spline data filtering. Linear and angular kinematic data of digitized points and the center of mass (COM) were recorded. The construction of key positions and 2D kinograms was developed by Poser software version 4.0.3.127 (1991–2000 Curius Labs^©^ Inc, Santa Cruz, CA, USA).

Before data collection, each participant performed a ten-minute warm-up including jogging, stretching, and jumping with stable landing exercises. Afterward, participants started in a standing position on the force plate, with 20 digital markers attached to their bodies. They were required to randomly [32] perform one of the SBS (i.e., SBS_270°_, SBS_180°_, and SBS_90°_) following a precise signal. To familiarize participants with the different arm-swing techniques, two to three attempts per technique were performed under the supervision of qualified judges. Afterward, two attempts were carried out for each of the SBS techniques (i.e., SBS_270°_, SBS_180°_, and SBS_90°_). The rest time between attempts was two minutes with five minutes allowed between the different techniques. Four experienced international judges marked all attempts. The best somersault of each arm-swing technique was retained for further analysis.

### 2.4. Statistical Analyses

Data are reported as mean ± standard deviation (SD) and 95% confidence intervals (95% CI). Effect size (d) was calculated using GPower software (Bonn FRG, Bonn University, Department of Psychology [33]). The following scale was used for the interpretation of d: <0.2, (trivial); 0.2–0.6, (small); 0.6–1.2, (moderate); 1.2–2.0, (large); and >2.0, (very large) [34,35]. The normality of the distribution, estimated by the Shapiro–Wilk test, was acceptable for all variables. Therefore, repeated measures ANOVA were applied to compare the different SBS arm-swing techniques. A pairwise comparison was conducted using Bonferroni post-hoc test. Additionally, the relative and absolute reliability of SBS (i.e., SBS_270°_, SBS_180°_, and SBS_90°_) were examined using the intra-class correlation coefficient (ICC) and the typical error of measurement (TEM) expressed as coefficient of variation (CV), respectively. The SWC was assumed by multiplying the between-subject SD by 0.2 (SWC_0.2_), indicating the typical small effect [36]. The ability of the test to detect a change was rated as “good”, “OK”, or “marginal” when the TEM was below, similar, or higher than the SWC_0.2_, respectively [37]. The minimal detectable change (MDC_95%_), which represents 95% CI of the difference in score between paired observations, was determined as MDC_95%_ = TEM · 1.96 · √2 [38]. The level of significance was set at *p* ≤ 0.05. Statistical analyses were performed using the software package SPSS 20.0 (SPSS, Inc., Chicago, IL, USA).

## 3. Results

The absolute and relative reliability of vertical ground reaction force (GRF) during landing across the three arm-swing modes (i.e., SBS_270°_, SBS_180°_, and SBS_90°_) was very high (Table 1).

Results of repeated measure ANOVA showed a significant difference in the kinetic and kinematic variables recorded across the three SBS arm-swing modes (Table 2). A pairwise comparison between the three execution modes (i.e., SBS_270°_, SBS_180°_, and SBS_90°_) is presented in Table 3. Results showed that compared to the other arm-swing techniques, the SBS_90°_ arm-swing resulted in the closest angle to the vertical. The same arm-swing technique resulted in the lowest horizontal and vertical force values upon landing compared to the other arm-swing techniques (SBS_270°_: *p* = 0.023 and 0.009, respectively; SBS_180°_: *p* = 0.004 and 0.080, respectively). Similar findings were observed for the horizontal velocity with the lowest values noted for the SBS_90°_ arm-swing technique (*p* = 0.021). However, the best opening angle was observed during the SBS_270°_ technique, since it presented the best vertical displacement.

An example of kinetics and kinematic characteristics of the three SBS execution modes (i.e., SBS_270°_, SBS_180°_, and SBS_90°_) is shown in Figure 2.

## 4. Discussion

This study aimed to analyze and compare the kinetic and kinematic parameters during the landing phase of SBS following three different arm-swing techniques performed during the preparatory phase in high-level male gymnasts. More specifically, we aimed to determine which of these three arm-swing techniques allows a more efficient and stable landing (stick-landing) and minimum joint stress. The main findings indicated that the SBS with a 90° arm-swing seems to favor a better absorption of the ground reaction force upon landing because it affords a lower impact intensity with the ground and a closer to the vertical landing angle compared with the other arm-swing techniques.

Two crucial criteria are considered when assessing the technical performance of SBS in gymnastics: vertical displacement of the gymnast’s COM and stable landing on the spot (i.e., stick-landing) without backward and/or lateral displacement. In fact, a better elevation of the COM promotes an early opening of the body (i.e., ungrouping) and affords faster and longer visual contact with the ground. This results in better stability during landing (i.e., less joint stress) especially when it is after a 360 degrees rotation like the back somersault [7]. The findings of this study showed that at the moment of landing, the COM angle of impact with the ground (i.e., when the feet touch the ground for the first time upon landing) varies significantly depending on the arm-swing technique used. The highest angle was observed during SBS_90°_. The SBS_90°_ landing angle was the closest to the vertical (∠_L_ = 77.63 ± 0.91°) compared to the other techniques (i.e., SBS_270°_ ∠_L_ = 75.60 ± 0.85° and SBS_180°_ ∠_L_ = 74.89 ± 1.20°). This seems to be due to a later ungrouping during the SBS_90°_. Hraski [39] compared seven different types of backward somersaults (i.e., tucked, picked, layout, layout with twist 366°, double tucked, double layout, and double tucked with twist 360° back somersault) in male elite gymnasts and demonstrated a landing angle of 76° from the vertical in tucked back somersault. Similarly, McNitt-Gray [40] examined the effects of speed on landing in male elite gymnasts and reported that a successful SBS landing angle is between 69° and 80°. Generally, a landing angle of more than 79.86° would lead to an over-rotation, and between 61° and 66° would result in a failed (i.e., unstable) landing.

Moreover, during the SBS_270°_, the thigh/trunk angle was higher (∠_H_ = 71.96 ± 0.75°) compared to the other arm-swing modes (i.e., SBS_180°_ ∠_H_ = 60.07 ± 8.08° and SBS_90°_ ∠_H_ = 56.53 ± 5.41°). This reflects an early opening compared to other conditions (i.e., SBS_180°_ and SBS_90°_). The present findings are in agreement with previous studies. For example, Beatty, et al. [41] compared different landings across different gymnastic skills (i.e., the straight jump; tucked jump; round-off; back flip; and the tucked back somersault) in sub-elite gymnasts. They reported a thigh/trunk angle of 77° at landing in the tucked back somersault similar to the SBS_270°_ execution mode. Sadowski et al. [42] analyzed basic acrobatic jump (i.e., tucked back somersault) in highly skilled male gymnastics acrobats and reported an ungrouping angle of 65° similar to the SBS_180°_ execution mode.

In terms of the kinetic parameters, our findings indicated that the impact of force on the ground upon landing varies significantly depending on the arm-swing technique used. In fact, we observed decreased horizontal and vertical force components during SBS_90°_ (F_x_ = 0.85 ± 0.27 BW and F_y_ = 9.92 ± 1.61 BW) compared to the other arm-swing techniques (i.e., SBS_270°_ F_x_ = 1.62 ± 0.34 BW and F_y_ = 13.94 ± 1.35 BW; SBS_180°_ F_x_ = 2.06 ± 0.32 BW and F_y_ = 12.69 ± 1.24 BW). This can be explained by an optimal elevation of the COM and a delayed opening with a landing angle close to the vertical during the SBS_90°_ compared to SBS_270°_ and SBS_180°_. In their studies, which examined mechanical loading and multi-joint control of the reaction force across different landings (i.e., drop landing and front and back tucked somersault) in male elite gymnasts, McNitt-Gray, Hester, Mathiyakom, and Munkasy [17] reported 0.82 BW in horizontal and 5.09 BW in vertical ground reaction force during the back tucked somersault. These results are lower than those obtained in our study. Furthermore, Wade, Campbell, Smith, Norcott, and O’Sullivan [27] examined the peak GRF at landing between different dynamic gymnastic skills (i.e., drop landing and front and back tucked somersault) in female elite gymnasts and reported 9.30 BW in vertical ground reaction force during the back tucked somersault. These results are similar to those collected during the SBS_90°_ execution mode. However, our results in the SBS_270°_ and the SBS_180°_ execution mode are higher than the above-cited works [17,27], but they are comparable with those of Cossin et al. [43] who reported 13.5 BW in vertical ground reaction force during back tucked somersault landing on three different Korean teeterboards in acrobat gymnasts.

In addition, the moment of force varies significantly (*p* = 0.000) across the three axes (i.e., M_z_, M_x_, and M_y_). The highest lateral and horizontal momentum (M_z_ = 17.80 ± 3.08 N·m^−1^ and M_x_ = 20.77 ± 6.97 N·m^−1^) was achieved during SBS_90°_ compared to the other conditions (i.e., SBS_270°_: M_z_ = 5.62 ± 1.58 N·m^−1^ and M_x_ = 17.86 ± 8.07 N·m^−1^; SBS_180°_: M_z_ = 11.09 ± 3.28 N·m^−1^ and M_x_ = 15.32 ± 4.06 N·m^−1^). This could be due to the landing angle and the delayed opening of the body, which could increase the lateral and horizontal momentum for better stabilization. However, the lowest vertical momentum was also observed during SBS_90°_ (M_y_ = 8.84 ± 0.74 N·m^−1^) compared to the other conditions (i.e., SBS_270°_: M_y_ = 10.95 ± 4.94 N·m^−1^ and SBS_180°_: M_y_ = 10.66 ± 2.43 N·m^−1^). This seems to be the result of an optimal COM vertical displacement (i.e., relatively small vertical displacement compared to other techniques) as well as a lower force impact with the ground. Further, our findings indicated that the horizontal velocity of the COM varies significantly between the three arm-swing techniques with SBS_90°_ presenting the lowest velocity (V_x_ = 0.038 ± 0.02 m·s^−1^). However, the lateral and vertical components of velocity remain relatively stable across the three arm-swing techniques.

For the power developed during landing, we noted different values depending on the arm-swing technique used. More specifically, SBS_270°_ induced the highest lateral and horizontal peak power upon landing (P_z_ = 0.56 ± 0.25 N·m·s^−1^ and P_x_ = 0.53 ± 0.21 N·m·s^−1^), compared to the other arm-swing techniques (i.e., SBS_180°_: P_z_ = 0.039 ± 0.02 N·m·s^−1^ and P_x_ = 0.040 ± 0.02 N·m·s^−1^; SBS_90°_: P_z_ = 0.043 ± 0.02 N·m·s^−1^ and P_x_ = 0.48 ± 0.29 N·m·s^−1^). The highest lateral and horizontal peak power upon landing, seen during SBS_270°_, may be due to the maximum lateral and horizontal force, as well as the vertical COM displacement offered by this arm-swing technique [7]. Nevertheless, the vertical peak power component upon landing remained relatively stable across the three arm-swing techniques.

## 5. Limitations and Future Research Perspectives

This study has some limitations that warrant discussion. First, the sample size is rather small. However, we recruited all the members of the male Tunisian national team. Additionally, unlike team sports, the overall population in AG is rather reduced, making the procedure of recruiting a large sample size very challenging, particularly at the elite level. Nevertheless, future studies with larger sample sizes are needed to reinforce the findings of the current investigation. Second, the analysis system used in this study could represent a limitation. This is because we used a semi-automatic system with just two high-speed cameras. Upcoming studies should favor using real-time motion analysis systems (e.g., Vicon).

## 6. Conclusions

The main results of this study showed that elite gymnasts experience different kinematic and kinetic characteristics upon landing depending on the arm-swing technique used. More specifically, the SBS_90°_ arm-swing technique (i.e., the Romanian school technique) seems to favor a better absorption of the ground reaction force by reducing the intensity of the impact with the ground and by allowing a landing angle closer to the vertical. Thus, the SBS_90°_ arm-swing technique results in less joint stress, which would probably lead to less injury. Practitioners could favor the SBS_90°_ arm-swing technique during the execution of SBS.

## Figures and Tables

**Figure 1 jfmk-08-00010-f001:**
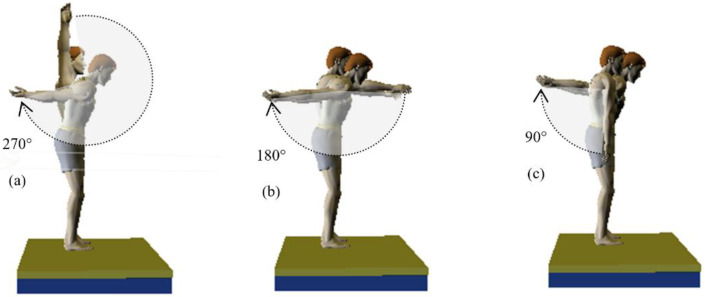
Technical arms swing during the preparatory phase of standing back somersault [23]. (**a**) Arm-swing technique with 270° backswing (SBS_270°_); (**b**) Arm-swing technique with 180° backswing (SBS_180°_); (**c**) Arm-swing technique with 90° backswing (SBS_90°_).

**Figure 2 jfmk-08-00010-f002:**
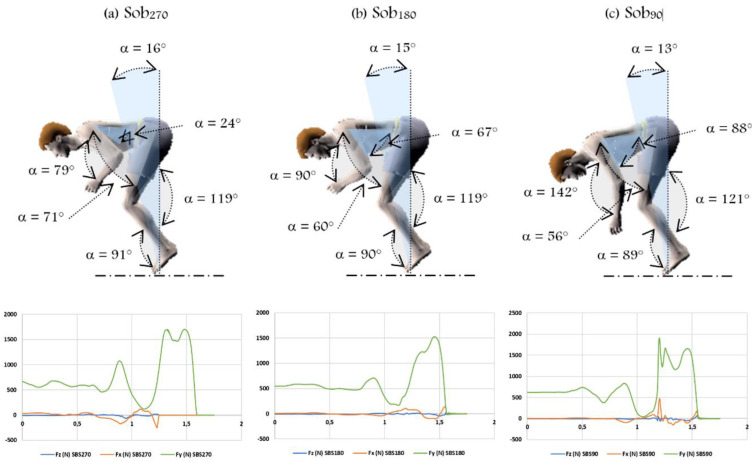
Examples of landing angles and ground reaction force during different standing back somersaults. (**a**) SBS_270°_; (**b**) SBS_180°_; (**c**) SBS_90°_.

**Table 1 jfmk-08-00010-t001:** Statistical analysis of the absolute and relative reliability of vertical ground reaction force measured during landing in standing back somersault.

R1 vs. R2	Mean ± SD F_y_ (N)	*T*-test (*p*)	TEM	TEM _(%)_	MDC _(95%)_	SWC _(0.2)_	ICC _(95% CI)_
SBS_270°_	7894.96 ± 1017.04 7908.05 ± 988.18	0.852	10.387	0.131	28.792	28.792	0.995 (0.901–0.999)
SBS_180°_	7182.43 ± 879.63 7251.65 ± 822.85	0.326	10.725	0.149	29.729	160.767	0.994 (0.880–0.999)
SBS_90°_	5555.94 ± 381.93 5561.18 ± 269.97	0.933	26.618	0.479	73.782	62.365	0.967 (0.443–0.996)

(R1) first repetition; (R2) second repetition; (F_y_) vertical ground reaction force; (TEM) typical error of measurement; (MDC) minimal detectable change; (SWC) smallest worthwhile change; (ICC) intra-class correlation coefficient.

**Table 2 jfmk-08-00010-t002:** ANOVA repeated measure between different standing back somersaults.

Variables		df	Mean Square	F	Sig.	Effect Size	Power
Kinetics	F_z_ (BW)	2	0.166	6.783	0.060	2.604 ^§^	0.507
	F_x_ (BW)	2	1.467	24.962	0.008	4.998 ^§^	0.955
	F_y_ (BW)	2	40.361	41.966	0.003	6.479 ^§^	0.996
	M_z_ (N·m^−1^)	2	130,748.283	104.189	0.000	10.203 ^§^	1.000
	M_x_ (N·m^−1^)	2	114,815.710	47.597	0.000	6.876 ^§^	1.000
	M_y_ (N·m^−1^)	2	13,944.128	149.906	0.000	12.241 ^§^	1.000
	P_z_ (N·m·s^−1^)	2	0.456	18.878	0.001	4.357 ^§^	0.996
	P^x^ (N·m·s^−1^)	2	0.368	10.260	0.006	3.199 ^§^	0.921
	P_y_ (N·m·s^−1^)	2	0.010	1.760	0.233	1.328 ^#^	0.267
Kinematics	V_z_ (m·s^−1^)	2	0.002	0.730	0.511	0.853 ^*^	0.134
	V_x_ (m·s^−1^)	2	0.004	8.388	0.011	2.895 ^§^	0.859
	V_y_ (m·s^−1^)	2	0.064	0.456	0.650	0.674 ^*^	0.101
	∠_L_ (°)	2	10.141	8.872	0.009	2.976 ^§^	0.878
	∠_H_ (°)	2	326.727	12.249	0.004	3.501 ^§^	0.958
	∠_K_ (°)	2	74.122	0.093	0.912	0.306 ^¤^	0.060

(F_z_) Lateral ground reaction force; (F_x_) Horizontal ground reaction force; (F_y_) Vertical ground reaction force; (M_z_) Lateral moment of force; (M_x_) Horizontal moment of force; (M_y_) Vertical moment of force; (P_z_) Lateral power; (P_x_) Horizontal power; (P_y_) Vertical power; (V_z_) Lateral velocity; (V_x_) Horizontal velocity; (V_y_) Vertical velocity; (∠_L_) Angle of landing; (∠_H_) Trunk legs angle; (∠_K_) Legs thighs angle; (^¤^) Small effect size; (^*^) Moderate effect size; (^#^) Large effect size; (BW) Body weight; (^§^) Very large effect size.

**Table 3 jfmk-08-00010-t003:** Pairwise comparisons Bonferroni Post-Hoc.

Measure		Mean Difference	Std. Error	Sig.	Effect Size
F_x_ (BW)	SBS_270°_ vs. SBS_180°_	−0.446	0.072	0.011	6.472 ^§^
	SBS_270°_ vs. SBS_90°_	0.766	0.153	0.023	5.006 ^§^
	SBS_180°_ vs. SBS_90°_	1.212	0.148	0.004	8.189 ^§^
F_y_ (BW)	SBS_270°_ vs. SBS_90°_	4.018	0.620	0.009	6.480 ^§^
M_z_ (N·m^−1^)	SBS_270°_ vs. SBS_180°_	281.433	26.824	0.001	10.491 ^§^
	SBS_270°_ vs. SBS_90°_	278.723	27.985	0.002	9.960 ^§^
M_x_ (N·m^−1^)	SBS_270°_ vs. SBS_180°_	277.346	33.018	0.003	8.399 ^§^
	SBS_180°_ vs. SBS_90°_	−244.497	21.986	0.001	11.120 ^§^
M_y_ (N·m^−1^)	SBS_270°_ vs. SBS_180°_	92.970	4.723	0.000	19.684 ^§^
	SBS_270°_ vs. SBS_90°_	89.889	8.115	0.001	10.076 ^§^
P_z_ (N·m·s^−1^)	SBS_270°_ vs. SBS_180°_	0.525	0.122	0.038	4.303 ^§^
	SBS_270°_ vs. SBS_90°_	0.521	0.117	0.034	4.452 ^§^
P_x_ (N·m·s^−1^)	SBS_180°_ vs. SBS_90°_	−0.494	0.103	0.026	4.796 ^§^
V_x_ (m·s^−1^)	SBS_180°_ vs. SBS_90°_	0.058	0.011	0.021	5.272 ^§^
∠_L_ (°)	SBS_270°_ vs. SBS_90°_	−2.028	0.475	0.039	4.269 ^§^
∠_H_ (°)	SBS_270°_ vs. SBS_90°_	15.434	2.508	0.011	6.153 ^§^

(F_x_) Horizontal ground reaction force; (F_y_) Vertical ground reaction force; (M_z_) Lateral moment of force; (M_x_) Horizontal moment of force; (M_y_) Vertical moment of force; (P_z_) Lateral power; (P_x_) Horizontal power; (V_x_) Horizontal velocity; (∠_L_) Angle of landing; (∠_H_) Trunk legs angle; (BW) Body weight; (^§^) Very large effect size.

## Data Availability

Data are available on request from the first author.

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
