# Peer review of "Kinetic and Kinematic Analysis of Landing during Standing Back Somersault Using Three Technical Arm Swings in Artistic Gymnastics"

_jfmk, 2023, doi:10.3390/jfmk8010010_

Round 1

Reviewer 1 Report

Thank you very much for the opportunity to review this manuscript. The topic of the paper is interesting and fits the scope of the journal. The text is relatively well written and composed. The major limitation of this study is the small sample size of control group, only 6 athletes. However, I have many minor comments that I believe that help to improve the paper. 

Introduction

Lines 57-58. Please replace “Mkaouer, Jemni, Amara, Chaabene, Padulo and Tabka [16]” with “Mkaouer et al. [16]”.

So, if I understand well the results of this study is closer to the Romanian school gymnasts.

Author Response

Dear Reviewer,

Thank you for your careful perusal of our manuscript and your constructive and helpful comments. We addressed all your concerns in our point-by-point statements and made changes to the manuscript whenever necessary. We believe that the changes introduced have improved the quality of the manuscript.  We hope that our manuscript adheres now to the publication standard of  Journal of Functional Morphology and Kinesiology.

Kind Regards,

Authors

COMMENTS AND SUGGESTIONS FOR AUTHORS

Comment #1: Thank you very much for the opportunity to review this manuscript. The topic of the paper is interesting and fits the scope of the journal. The text is relatively well written and composed. The major limitation of this study is the small sample size of control group, only 6 athletes. However, I have many minor comments that I believe that help to improve the paper.

Authors’ response to comment #1: Thank you for your positive feedback. Highly appreciated.

INTRODUCTION

Comment #2: Lines 57-58. Please replace “Mkaouer, Jemni, Amara, Chaabene, Padulo and Tabka [16]” with “Mkaouer et al. [16]”.

 Authors’ response to comment #2: Done as suggested. Thank you.

Comment #3: So, if I understand well the results of this study is closer to the Romanian school gymnasts.

 Authors’ response to comment #2: Thank you for your comment. Our results showed that the Romanian school technique seems to be the most adequate. The SBS90 arm-swing technique (i.e., Romanian school) results in less joint stress, which would probably lead to less injury.

Reviewer 2 Report

GENERAL COMMENTS

The aim of this paper was to examine the kinetic and kinematic parameters during the landing phase of standing back somersaults following three technical arm swings performed during the preparatory phase in high-level male gymnasts. Although this article addresses some interesting findings, many issues should be addressed before publication.

SPECIFIC COMMENTS

ABSTRACT

Please state the actual p-value for the results.

INTRODUCTION
The introduction needs minor revision and clarification. First, the aim of the study is not clear. The research manuscript should answer a specific research question lacking in this study. Also, the way the authors build up their introduction does not lead to the research question. Although much of the necessary information regarding the background is already briefly written down, the authors should re-structure their introduction, explaining why their research is important. Why the measurements used are good measures. Why use the three-arm swing technique? Is it a valid measure? The authors described some studies separately, but it is very shallow and not focused. Thus, it is recommended that the authors expand this part, which should lead to a clear research question.

METHODS
The methods section needs major revision. I am very confused.

Firstly, I checked the reference for sample size calculation but could not get 6 participants. I calculated, and it required 28 participants. Please show the calculation and why the effect size is so high? It seemed like the entire study was compromised as the power of the study was not enough.

2. Please provide an ethical approval code when dealing with human participants.

3. In the introduction, it mentioned comparing Russian, Chinese and Romanian but in the manuscript only Tunisia?

4. The procedures for collecting data is too brief, please elaborate.

5. Please state the protocol for the kinetic and kinematic measures. How was measured?

In my opinion, the authors should justify why only 6 participants. Besides, with only 6 participants, I don’t think that authors can use repeated measure ANOVA as it does not meet the criteria, which will create the type-2 error.

RESULTS
The results section needs major revision. The authors should add how they gathered all the information. Probably, including a research question would help the authors to structure their results.
DISCUSSION
In the discussion section, the authors should further discuss their findings and the implication of these findings. They should also discuss their findings in more depth. In addition, they describe many studies in great detail, which is not necessary for the discussion. I think the authors should also add limitations to this study. Please revise.

Thank you.

Author Response

Dear Reviewer,

Thank you for your careful perusal of our manuscript and your constructive and helpful comments. We addressed all your concerns in our point-by-point statements and made changes to the manuscript whenever necessary. We believe that the changes introduced have improved the quality of the manuscript.  We hope that our manuscript adheres now to the publication standard of the Journal of Functional Morphology and Kinesiology.

Kind Regards,

Authors

Reviewer’s 2 comments (Changes in the main text are highlighted in blue)

 GENERAL COMMENTS

 The aim of this paper was to examine the kinetic and kinematic parameters during the landing phase of standing back somersaults following three technical arm swings performed during the preparatory phase in high-level male gymnasts. Although this article addresses some interesting findings, many issues should be addressed before publication.

SPECIFIC COMMENTS

ABSTRACT

 Comment #1: Please state the actual p-value for the results.

Authors’ response comment #1: Thank for your comment. We have included the actual p-value as suggested.

 INTRODUCTION

Reviewer’s comment #2: The introduction needs minor revision and clarification. First, the aim of the study is not clear. The research manuscript should answer a specific research question lacking in this study. Also, the way the authors build up their introduction does not lead to the research question. Although much of the necessary information regarding the background is already briefly written down, the authors should restructure their introduction, explaining why their research is important. Why the measurements used are good measures. Why use the three-arm swing technique? Is it a valid measure? The authors described some studies separately, but it is very shallow and not focused. Thus, it is recommended that the authors expand this part, which should lead to a clear research question.

Authors’ response to Reviewer’s comment #2: Thank you for your helpful comments and suggestions. We substantially revised the structure and the content of the introduction. We also worked on highlighting the deficit in the literature and therefore the need for this study. We particularly emphasized the relevance of this study from practical and health (i.e., injury prevention) perspectives. We believe that the introduction has significantly improved and we would be happy to make further changes in a second round of revisions if deemed necessary. The revised introduction reads as follows:

“Artistic gymnastics (AG) is a judgment sport in which the landing concludes every event/acrobatic series [1]. As such, the final judgment score is highly determined by the quality of landing [1]. Earlier studies showed a low success rate at landing in AG, with a high error rate reaching 71.9 % on floor exercise [2]. There is also evidence indicating that the rate of lower limb injuries is high during the landing phase on the floor exercise (40%) [3-6].

On the other hand, the preparation phase (i.e., run-up technique), such as during the floor exercise, is crucial for the success and safety of acrobatic elements in AG. In fact, the preparation phase moderates the amount of momentum necessary for the successful performance of acrobatic elements [7]. More particularly, it has been shown that the preparation phase of an acrobatic element could affect its last phase (i.e., landing phase) [8]. In this sense, the actions of the arms, such as during the preparation phase are very important for a successful performance of acrobatic movements during the floor exercise [7,9,10]. In this sense, there is evidence that the inertia generated by the displacement, speed, and/or opening angle of the arms can markedly affect the quality of the acrobatic element as a whole [7,11] and the landing phase more particularly [8].

The standing back somersault (SBS) is a common floor acrobatic exercise. The preparation phase of the SBS can be performed according to three different schools of AG (i.e., Chinese, Romanian, and Russian) [7]. The difference between these schools is mainly related to the action of the arms. In the Russian school, gymnasts start the arms vertically (arm/trunk angle opening is 180°). Then perform a 270° oscillation in the descending phase by lowering the arms from the top to the front and the back to the end in the back-ward horizontal position [7,12-15], (Figure 1a). According to the Chinese school, gymnasts begin with arms extended horizontally forward; in the downward phase, they then perform a 180° oscillation by lowering them downwards and backwards to reach a horizontal back position [7,16-19], (Figure 1b). However, as per the Romanian school, gymnasts start with their arms extended and lowered along the body; they perform a 90° oscillation going backwards, ending in a horizontal back position [7,20-22], (Figure 1c).

It is worth noting that several previous studies focused on the landing phase during the SBS [17,24-28]. However, none of them has addressed the effects of different technical arm swings during the preparation phase on the final phase, which is the landing phase of the SBS. Therefore, this study aimed to compare the kinetic and kinematic parameters during the landing phase of SBS following three different technical arm swings performed during the preparatory phase in high-level male gymnasts. More specifically, we aimed to identify which of the three arm swing techniques during the preparatory phase results in a more efficient and stable landing (stick-landing) with a minimum joint impact.”

METHODS

Reviewer’s comment #3: The methods section needs major revision. I am very confused.

Firstly, I checked the reference for sample size calculation but could not get 6 participants. I calculated, and it required 28 participants. Please show the calculation and why the effect size is so high? It seemed like the entire study was compromised as the power of the study was not enough.

Authors’ response to Reviewer’s comment #3: We thank the reviewer for his/her pertinent remark. Apologies for the confusion. We added more details related to the a priori power analysis. We hope that this aspect is clearer now and we are happy to discuss this with the reviewer in a second round. The revised version reads as follows:  

“A priori power analysis with type I error of 0.05 and 80% statistical power was computed using GPower software (Version 3.1, University of Dusseldorf, Germany [29]). The analysis indicated that a minimum of 6 participants are sufficient to observe a significant, large effect size (f =50) for kinetic (i.e., vertical and horizontal ground reaction force) and kinematic variables (i.e., joint angles and velocity) [7]. Therefore, six senior men’s artistic gymnastics members of the Tunisian national team (age 23.17±1.61 years; height 1.65±0.05 m; body mass 56.80±7.66 kg) agreed to participate in this study. Of note, all gymnasts are familiar with the three arm-swing techniques as they have been trained by Romanian, Russian, and Chinese coaches throughout their careers.”

All the details related to the calculations are as follows:  

F tests - ANOVA: Repeated measures, within factors

Analysis:   A priori: Compute required sample size

Input:        Effect size f                            =  0.50

                   α err prob                                =  0.05

                   Power (1-β err prob)              =  0.80

                   Number of groups                  =  1

                   Number of measurements      =  6

                   Corr among rep measures      =  0.5

                   Nonsphericity correction ε     =  1

Output:     Noncentrality parameter λ     =  18.0000000

                   Critical F                                =  2.6029874

                   Numerator df                          =  5.0000000

                   Denominator df                      =  25.0000000

                   Total sample size                   =  6

Converting Cohen’s “d” of 1.00 to “f” gives 0.50

Reviewer’s comment #4: Please provide an ethical approval code when dealing with human participants.

Authors’ response to Reviewer’s comment #4: More details related to the ethical approval were included in the revised version as follows:

“The experimental protocol was performed per the Declaration of Helsinki for human experimentation [43] and was approved by the Ethical Committee of the National Centre of Medicine and Science in Sport (LR09SEP01)”. Thank you.

Reviewer’s comment #5: In the introduction, it mentioned comparing Russian, Chinese and Romanian but in the manuscript only Tunisia?

Authors’ response to Reviewer’s comment #5: Thank you for your comment. Please note that this study aimed to contrast three different arm swing techniques that belong to three different artistic gymnastic schools during the execution of the standing back somersault exercise. These are the Russian, Chinese, and Romanian schools. The included participants were members of the Tunisian artistic gymnastics national team. We have clarified, however, that all gymnasts are familiar with the three arm-swing techniques as they have been trained by Romanian, Russian, and Chinese coaches throughout their careers. The following changes were included in the revised version of the manuscript:

“Therefore, six senior men’s artistic gymnastics members of the Tunisian national team (age 23.17±1.61 years; height 1.65±0.05 m; body mass 56.80±7.66 kg) agreed to participate in this study. Of note, all gymnasts are familiar with the three arm-swing techniques as they have been trained by Romanian, Russian, and Chinese coaches throughout their careers..”

Reviewer’s comment #6: The procedures for collecting data is too brief, please elaborate.

Authors’ response to Reviewer’s comment #6: Thank you for your comment. More details were added to the experimental procedures as follows:

“The experiment took place in the laboratory of the National Centre of Medicine and Science in Sport across three days, starting at 4:00 PM to 6:00 PM under the following environmental conditions: average temperature of 23°C and humidity of 35%. Two Kistler force plates were synchronized with two NAC high-speed cameras. Both cameras were placed at 5m from the center of the force plate, the first one at the front and the second sideways. The peak force and the moment of force (i.e., quantified as the peak slope of the force–time curve [i.e., Δforce/Δtime]), upon landing during the SBS were recorded. The kinematic analysis was conducted in three-dimension (3D). A semi-automatic digitalization was used with quantic-spline data filtering. Linear and angular kinematic data of digitized points and the center of mass (COM) were recorded. The construction of key positions and 2D kinograms was developed by Poser software version 4.0.3.127 (1991–2000 Curius Labs© Inc).

Before data collection, each participant performed ten minutes warm-up including jogging, stretching, and jumping with stable landing exercises. Afterward, participants started in a standing position on the force plate, with 20 digital markers attached to their bodies. They were required to randomly [32] perform one of the SBS (i.e., SBS270°, SBS180° and SBS90°) following a precise signal. To familiarize participants with the different arm-swing techniques, two to three attempts per technique were performed under the supervision of qualified judges. Afterward, two attempts were carried out for each of the SBS techniques (i.e., SBS270°, SBS180° and SBS90°). The rest time between attempts was two minutes with five minutes allowed between the different techniques. Four experienced international judges marked all attempts. The best somersault of each arm-swing technique was retained for further analysis.”

Reviewer’s comment #7: Please state the protocol for the kinetic and kinematic measures. How was measured?

Authors’ response to Reviewer’s comment #7: Thank you for your comment. Please refer to our answer to comment 6.

Reviewer’s comment #8: In my opinion, the authors should justify why only 6 participants. Besides, with only 6 participants, I don’t think that authors can use repeated measure ANOVA as it does not meet the criteria, which will create the type-2 error.

Authors’ response to Reviewer’s comment #8: Thank you for your pertinent comments. We included more details related to the a priori power analysis as mentioned in our answer to comment 3. Please note that the effect size used to estimate sample size was based on a previous similar study by Mkaouer et al. (2014) (full reference below). However, we agree with the reviewer that our sample size is rather small. That’s why we have included this specific point among the limitations of this study. It is worth noting though that we recruited all the members of the male Tunisian national team. Nevertheless, unlike team sports, the overall population in artistic gymnastics is rather reduced, making the procedure of recruiting a larger sample size very challenging. Therefore, future studies including more number of participants are indeed needed to confirm the results of the present study. In terms of the statistical approach used, we have checked the normality assumption using the Shapiro-Wilk test and data was normally distributed. As such, ANOVA was deemed appropriate for data analysis. To double-check the main findings, we have used the non-parametric equivalence (i.e., Friedman’s test) of ANOVA and we obtained the same results.

Mkaouer, B.; Jemni, M.; Amara, S.; Chaabene, H.; Padulo, J.; Tabka, Z. Effect of three technical arms swings on the elevation of the center of mass during a standing back somersault. Journal of human kinetics 2014, 40, 37-48, doi:0.2478/hukin-2014-0005.

RESULTS

Reviewer’s comment #9: The results section needs major revision. The authors should add how they gathered all the information. Probably, including a research question would help the authors to structure their results.

Authors’ response to Reviewer’s comment #9: Thank you for your comment. We revised the results section by adding a more detailed interpretation of the findings as follows:

“Results of repeated measure ANOVA showed a significant difference in the kinetic and kinematic variables recorded across the three SBS arm-swing modes (Table 2). A pairwise comparison between the three execution modes (i.e., SBS270°, SBS180° and SBS90°) is presented in Table 3. Results showed that compared to the other arm-swing techniques, the SBS90° arm-swing resulted in the closest angle to the vertical. The same arm-swing technique resulted in the lowest horizontal and vertical force values upon landing compared to the other arm-swing techniques (SBS270°: p=0.023 and 0.009, respectively; SBS180°: p=0.004 and 0.080, respectively). Similar findings were observed for the horizontal velocity with the lowest values noted for the SBS90° arm-swing technique (p=0.021). However, the best opening angle was observed during the SBS270° technique, since it presented the best vertical displacement.”

DISCUSSION

Reviewer’s comment #10: In the discussion section, the authors should further discuss their findings and the implication of these findings. They should also discuss their findings in more depth. In addition, they describe many studies in great detail, which is not necessary for the discussion. I think the authors should also add limitations to this study. Please revise.

Authors’ response to Reviewer’s comment #10: Thank you for your legitimate comments. The discussion section was substantially revised accordingly. Additionally, we added a limitation section in line with the reviewer’s suggestion as follows:

“This study has some limitations that warrant discussion. First, the sample size is rather small. However, we recruited all the members of the male Tunisian national team. Additionally, unlike team sports, the overall population in AG is rather reduced, making the procedure of recruiting a large sample size very challenging, particularly at the elite level. Nevertheless, future studies with larger sample sizes are needed to reinforce the findings of the current investigation. Second, the analysis system used in this study could represent a limitation. This is because we used a semi-automatic system with just two high-speed cameras. Upcoming studies should favor using real-time motion analysis systems (e.g., Vicon).”

Round 2

Reviewer 2 Report

I really appreciate the handwork from the authors for amending the manuscript point-by-point. I am satisfied with all the corrections done, thus recommend accepting this manuscript. All the best.